# Biostimulant Activity of Silicate Compounds and Antagonistic Bacteria on Physiological Growth Enhancement and Resistance of Banana to *Fusarium* Wilt Disease

**DOI:** 10.3390/plants12051124

**Published:** 2023-03-02

**Authors:** Md Aiman Takrim Zakaria, Siti Zaharah Sakimin, Mohd Razi Ismail, Khairulmazmi Ahmad, Susilawati Kasim, Ali Baghdadi

**Affiliations:** 1Department of Crop Science, Faculty of Agriculture, Universiti Putra Malaysia, Serdang 43400, Selangor, Malaysia; 2Institute of Tropical Agriculture and Food Security (ITAFoS), Universiti Putra Malaysia, Serdang 43400, Selangor, Malaysia; 3Department of Plant Protection, Faculty of Agriculture, Universiti Putra Malaysia, Serdang 43400, Selangor, Malaysia; 4Department of Land Management, Faculty of Agriculture, Universiti Putra Malaysia, Serdang 43400, Selangor, Malaysia; 5Department of Agricultural and Food Sciences (DISTAL), University of Bologna, 40127 Bologna, Italy

**Keywords:** *Fusarium*, banana, silicate, biostimulant, antagonistic bacteria

## Abstract

Biostimulants such as silicate (SiO_3_^2−^) compounds and antagonistic bacteria can alter soil microbial communities and enhance plant resistance to the pathogens and *Fusarium oxysporum f.* sp. *cubense* (FOC), the causal agent of *Fusarium* wilt disease in bananas. A study was conducted to investigate the biostimulating effects of SiO_3_^2−^ compounds and antagonistic bacteria on plant growth and resistance of the banana to *Fusarium* wilt disease. Two separate experiments with a similar experimental setup were conducted at the University of Putra Malaysia (UPM), Selangor. Both experiments were arranged in a split-plot randomized complete block design (RCBD) with four replicates. SiO_3_^2−^ compounds were prepared at a constant concentration of 1%. Potassium silicate (K_2_SiO_3_) was applied on soil uninoculated with FOC, and sodium silicate (Na_2_SiO_3_) was applied to FOC-contaminated soil before integrating with antagonistic bacteria; without *Bacillus* spp. ((0B)—control), *Bacillus subtilis* (BS), and *Bacillus thuringiensis* (BT). Four levels of application volume of SiO_3_^2−^ compounds [0, 20, 40, 60 mL) were used. Results showed that the integration of SiO_3_^2−^ compounds with BS (10^8^ CFU mL^−1^) enhanced the physiological growth performance of bananas. Soil application of 28.86 mL of K_2_SiO_3_ with BS enhanced the height of the pseudo-stem by 27.91 cm. Application of Na_2_SiO_3_ and BS significantly reduced the *Fusarium* wilt incidence in bananas by 56.25%. However, it was recommended that infected roots of bananas should be treated with 17.36 mL of Na_2_SiO_3_ with BS to stimulate better growth performance.

## 1. Introduction

In Malaysia, the banana is well-known as Pisang, with a total production of about 318,518 metric tons of fresh banana fruits [1]. However, hot and humid weather brought on by climate change hastens the onset and incidence of disease [2]. Stunted root and shoot growth performance caused by disease infestation resulted in reduced banana production and farmers’ decreased revenue. According to Wong et al. [3], *Fusarium* wilt disease caused by *Fusarium oxysporum* f. sp. *cubense* (FOC) affected more than 70% of edible banana cultivars in nine outbreak states in Peninsular Malaysia, including Penang, Perak, Selangor, Negeri Sembilan, Melaka, Johor, Pahang, Kelantan, and Terengganu. According to Staver et al. [4], the spread of *Fusarium* wilt disease could be categorized as one of the most destructive banana diseases in history that has been projected to reach 17% of the global banana growing area by 2040 equaling 36 million tons of products worth over USD 10 billion. Many factors affect the spread of *Fusarium* wilt in bananas growing in field conditions, including meteorological factors (rainfall, temperature, and cyclonic weather) and soil factors (soil type, suppressive soils, soil water content, and soil aeration) [5]. Currently, farmers rely heavily on synthetic fungicides to prevent yield losses caused by fungal diseases, but the extensive use has resulted in the emergence of fungicide-resistant pathogens, raising worries about the residual effects on food, the environment, and human health [6]. Nowadays, many researchers are looking for new plant biostimulants to enhance yield production apart from disease suppression. So, new environmentally safe and cost-effective methods utilizing plant defence inducers and plant growth-promoting rhizobacteria were developed for controlling soil-borne diseases without leaving residue in the fruit [7].

According to Gbongue et al. [8], SiO_3_^2−^ compound treatment is considered as an eco-friendly approach that has great potential as a plant biostimulant to enhance growth performance and systemic defence mechanisms against pathogens in the banana plantation. Deposition of SiO_3_^2−^ compound inhibits FOC from passing through the epidermal cell wall and provides the host plant with nutrients and resilience. Using antagonist bacteria against infections is one of the ways to inhibit the proliferation of plant diseases as well as stimulate plant growth development [9]. Based on previous findings, *Bacillus* strains effectively suppress fungal pathogens, and it also has the ability to produce more indole 3-acetic acid (IAA) hormone to regulate cell elongation, vascular tissue development, and apical dominance of the plant host [10,11]. Ezrari et al. (2022) [12] found that the application of BS together with Si-based compound significantly reduced mycelial growth of pathogenic *Fusarium* and suppressed dry root rot disease in citrus. Farhat et al. (2018) [13] also reported that the efficacy of SiO_3_^2−^ compounds was significantly increased by antagonistic bacteria and fungi inoculation for powdery mildew disease suppression in wheat plants. Ahmad et al. (2021) [14] revealed that SiO_3_^2−^ compounds applied on the root of bean plants grown under stress conditions significantly increased Chl_a+b_ content, leaf area, and enhanced Ps efficiency, which resulted in better morphology of growth performance and higher biomass production. Integration of SiO_3_^2−^ compounds with BS and *Bacillus amyloliquefaciens* enhanced plant growth development and leaf gas exchange in bananas. 

Plants treated with BS had the greatest total dry biomass production and managed to stimulate early and better photosynthetic activity [15]. Din et al. (2018) [16] stated that the growth of infected banana plants had been involved in physiological stress associated with a decrease in the photosynthetic performance simultaneously decrease transpiration rate of chlorotic leaves. Zellner et al. (2021) [17] also mentioned that banana plants infected by pathogens without being treated with Si clearly showed early leaf chlorosis that obviously affected their physiological growth activity. Thus, in the present investigation, attempts were made to evaluate the efficacy of integrating silicate compounds and the antagonist from *Bacillus* spp. in managing *Fusarium* wilt in bananas.

## 2. Results

### 2.1. Total Chlorophyll Content and Leaf Gas Exchange

Chlorophyll content in the leaf tissues is one of the most critical elements in regulating photosynthetic capacity and physiological performance as well. As shown in Figure 1, Chl_a+b_ content in the banana leaves is noticeably affected when antagonistic bacteria are added, with significant interactions between treatments (*p* < 0.05). Indeed, exogenous application of K_2_SiO_3_ integrated with different species of antagonistic bacteria on FOC-uninoculated planting media significantly increased in Chl_a+b_ content by 25.90% (BS), 13.45% (BT) relative to the 0B (4.88 mgcm^−2^) which served as negative control (−ve FOC). The amount of Chl_a+b_ content in the Berangan leaves seedling was significantly reduced when the roots of banana seedlings were infected with FOC. The Chl_a+b_ content in the infected plants (+ve FOC) showed significant decrease of 22.48% (0B) at 12 weeks after transplanting (WAT) compared with the negative control plants (−ve FOC). Application of SiO_3_^2−^ compounds significantly increased the Chl_a+b_ content in the Berangan leaves; the highest increase in the amount of Chl_a+b_ of 58.80% was gained when integrated with BS, followed by 48.18% for BT, in comparison with the 0B, which gained the lowest amount of Chl_a+b_ content (3.86 mgcm^−2^). 

Application of SiO_3_^2−^ compounds integrated with antagonistic bacteria significantly (*p* < 0.05) impacted the leaf gas exchange processes, especially the photosynthesis rate (Ps) and stomatal conductance; however, there was no significant interaction between these factors (Table 1). At six WAT, the Ps was significantly reduced (by 20.83%), with rising *Fusarium* stress infection, compared with the positive control (+ve FOC + 0B); somehow, there was no significant effect on stomatal conductance in both treatments applied on soil planting media and roots of banana. In addition, the integration of Na_2_SiO_3_ with antagonistic bacteria in the root treatments of banana seedlings also significantly affected the transpiration rate from the third leaf (*p* < 0.05). In line with this, there was a significant interaction between these factors based on the variance analysis, and the result is presented in Figure 2. It shows that the transpiration rate for negative control seedling (−ve FOC + 0B) significantly increased (by 40.71%) compared with the positive control (+ve FOC + 0B).

The stomatal conductance significantly increased, 21.81% (BS) and 34.54% (BT), compared with the negative control (−ve FOC + 0B), gaining 0.55 mmol m^−2^s^−1^. Application of Na_2_SiO_3_ with antagonist bacteria on infected soil planting media significantly increased Ps, 10.51% for BS and 10.83% for BT, in comparison with the negative control (15.41 µmol CO_2_ m^−2^s^−1^). Despite these findings, the transpiration rate from a third fully expanded leaf from BT-inoculated banana plants was significantly reduced, while the transpiration rate from 0B (3.93 mmol H_2_O m^−2^s^−1^), serving as a positive control (+ve FOC), increased by 0.76%. Application Na_2_SiO_3_ with different antagonistic bacteria under *Fusarium* stress condition significantly increased the process of transpiration rate, 27.46% (BS) and followed by 72.53% (BT) compared with the negative control (−ve FOC + 0B).

### 2.2. Physiological Attribute and Biochemical Changes

From the interaction effects in Figure 3, results showed that the increased water-use efficiency (WUE) and intrinsic WUE (Int-WUE) of banana seedlings were significantly increased (*p* < 0.05) when SiO_3_^2−^ compounds were applied by drenching the soil with antagonistic bacteria inoculation in the polybag. Interestingly, soil application with K_2_SiO_3_ integrated with BS significantly increased the WUE and Int-WUE, by 6.76% and 0.28%, whereas integration of K_2_SiO_3_ and BT significantly increased the WUE, by 5.02% and decreased the Int-WUE by 0.84% compared with the negative control (−ve FOC + 0B). Variance analysis results of studied traits in Table 2 showed that the proline content in the banana leaves was significantly increased, by 24.71%, but the accumulation of total flavonoid content (TFC), total phenolic content (TPC), and lignin content from the sampled roots of banana seedlings under *Fusarium* stress condition was significantly increased, by 39.65%, 29.12%, and 33.33%, respectively, compared with the positive control (+ve FOC + 0B). 

Exogenous application of Na_2_SiO_3_ together with antagonistic bacteria on the root of banana treated under *Fusarium* stress condition significantly decreased proline content, TFC, and TPC, by 12.62%, 37.09%, and 13.86%, respectively, but increased the total lignin content by 73.39% when inoculated with BS, compared with the 0B. However, inoculation with BT significantly increased the proline content by 11.54%, TPC by 8.57%, and the total lignin content in banana root tissues by 51.37%, even though the TFC decreased by 14.80% when BT was inoculated on planting media, compared with the control (0B). There was a significant increase of 20.74% in WUE and 47.70% in Int-WUE when bananas were infected with *Fusarium* wilt disease in comparison with the negative control (−ve FOC + 0B). However, integration of Na_2_SiO_3_ with BS under biotic stress significantly reduced the WUE and Int-WUE by 3.79% and 24.75%, respectively. Planting media inoculated with BT significantly increased the WUE by 28.02% and decreased the Int-WUE by 38.45%.

### 2.3. Disease Assessment

Based on the line graph in Figure 4, the negative control (−ve FOC + 0B) did not display *Fusarium* wilt symptoms till the end of the experiment. On the other hand, bananas grown on soil planting media challenged with FOC showed foliar symptoms of *Fusarium* wilt disease by a general yellowing of the older leaves and then wilting. The gradual development of disease symptoms was observed in the effects on the efficacy of Na_2_SiO_3_ combined with different species of antagonist bacteria (0B, BS, and BT) under the *Fusarium* stress condition. From these findings, a lower disease incidence (DI) indicates disease suppression by effective antagonistic bacteria combined with Na_2_SiO_3_ treatments. DI in treatments first and second week after FOC inoculation was 0%. The disease progression in Berangan banana was gradually increased with a DI rise of 45.83% observed at 12th WAT. 

These results recommended that the application of Na_2_SiO_3_ on the root of banana plant pre-inoculated with FOC significantly reduced DI, 56.25% (BS) and 37.5% (BT), compared with the negative control (−ve FOC + 0B), which gained the highest DI increase of 83.33% throughout the 12th WAT of assessment. Consequently, there was no significant difference in the population of bacteria in both treatments of planting media at the final phase of the experiment (Table 3). In contrast to the soil without the application of antagonistic bacteria (0B), BS-inoculated soil exhibited a decrease of 8.20% in the fungal population, whereas the bacteria population increased by 18.39%. BT dramatically reduced the fungus population by 4.45%, but it significantly boosted the bacteria population by 16.42% when compared with positive control (+ve FOC + 0B). However, during the final phase, the fungus population was 3.15% higher in the sampled soil infected by FOC (+ve FOC) compared with the negative control (−ve FOC + 0B). This is due to the introduction of FOC into the soil as well as the presence of indigenous fungus in the soil. Therefore, the survival of the soil microbial population was affected by the integration of SiO_3_^2−^ compounds with different species of antagonist bacteria.

### 2.4. Crop Growth Performance

Figure 5 shows that there was a significant increase of 62.11%, 34.66%, and 30.74% in the total leaf area for the 20 mL, 40 mL, and 60 mL of Na_2_SiO_3_, respectively, compared with 0 mL (1046.21 cm^2^) in the negative control (−ve FOC). Total leaf area was markedly affected by the *Fusarium* stress condition, in which the reduction was 11.36% in control inoculated with FOC, equal to 939.43 cm^2^ at 9^th^ WAT, compared with the control uninoculated with FOC. However, there were numerical differences between the treatments application of Na_2_SiO_3_ under *Fusarium* stress condition with a significant increase of 42.13% (20 mL), 17.14% (40 mL) and 7.24% (60 mL) in total leaf area of Berangan seedling, compared with 0 mL which served as control inoculated with FOC (939.43 cm^2^). 

The total dry biomass presented in Figure 6 shows a significantly high interaction between treatments of banana seedlings and the application volume of SiO_3_^2−^ compounds. Application volume of SiO_3_^2−^ compounds exhibited significantly different total dry biomass at 6th to 12th WAT but were not affected at 3th WAT. Total dry biomass harvested from plants treated by Na_2_SiO_3_, 12th WAT was significantly increased by 35.02% (20 mL), 26.59% (40 mL), and 4.33% (60 mL), compared with 0 mL serving as the FOC-uninoculated control (34.86 g). On the adversative effect of *Fusarium* stress condition after 12th WAT, there was a substantial reduction of 23.06% in control inoculated with FOC equal to 26.82 g, compared with the negative control (−ve FOC). However, the application volume of Na_2_SiO_3_ under *Fusarium* stress condition significantly influenced the total dry biomass, thus indicating that the average total dry biomass 12th WAT was 20 mL, which had the highest total dry biomass (44.23 g), followed by 40 mL (39.08 g) and 60 mL (32.36 g), while 0 mL serving as control uninoculated with FOC showed the lowest total dry biomass with the mean value of 26.82 g. 

Moreover, dry biomass distribution to specific organs is used to evaluate shoot and root growth performance based on the root-to-shoot ratio. Figure 7 shows the application of NaSiO_3_ at 9th WAT, with 40 mL gaining the highest value of 0.309, followed by 60 mL (0.288) and 20 mL (0.253). The 0 mL serving as the control uninoculated with FOC had the lowest value of 0.170. In contrast, when the planting media of banana seedlings inoculated with antagonistic bacteria were tested together with an enriched Si under *Fusarium* stress conditions and significantly improved the root-to-shoot ratio at 9th WAT, there was 20 mL of Na_2_SiO_3_ gained that had the highest value of 0.195, followed by 40 mL (0.150), and 60 mL (0.076), whereas 0 mL had the lowest root-to-shoot ratio of 0.06.

Figure 8 demonstrates that the SLA significantly increased by 17.22% (20 mL), 18.53% (40 mL), and 18.45% (60 mL) when Na_2_SiO_3_ was applied on banana seedling, compared with 0 mL which served as control uninoculated with FOC (53.36 cm^2^g^−1^). The SLA of Berangan seedling was significantly decreased by 6.09% with rising *Fusarium* stress condition, compared with the negative control (−ve FOC). The result of SLA is shown, thus indicating that the average SLA of Berangan banana seedlings 9th WAT was 0 mL of Na_2_SiO_3,_ which had the highest SLA (50.11 cm^2^g^−1^), followed by 40 mL (49.00 cm^2^g^−1^), and 20 mL (45.55 cm^2^g^−1^), while 60 mL gained the lowest SLA (41.99 cm^2^g^−1^). 

The bar chart in Figure 9 demonstrates a noticeable relative growth rate (RGR) with very significant interactions with inconsistent RGR results in Berangan bananas, especially seedlings grown under *Fusarium* stress condition. Obviously, different application volumes of SiO_3_^2−^ compounds exhibited a highly significant difference in the RGR 3th to 12th WAT, but they were not affected at 2th WAT. The RGR from the 6th to the 9th WAT was higher when the plants were treated with 20 > 0 > 40 > 60 mL of Na_2_SiO_3_. Moreover, the growth performance of bananas was significantly higher in RGR within the 9th to 12th WAT when seedlings were treated with 40 < 0 < 20 < 60 mL of Na_2_SiO_3_ and grown without *Fusarium* stress condition (−ve FOC). Thus, plant biostimulants based on SiO_3_^2−^ compounds integrated with antagonist bacteria offer a new strategy for the banana plant in enhancing plant growth performance and disease suppression. 

## 3. Discussion

Both experiments investigated the biostimulating effects of Si-based compounds and antagonistic bacteria on the integrated *Fusarium* wilt disease suppression to enhance banana growth performance. As anticipated, based on the previous findings, SiO_3_^2−^ compounds integrated with BS were shown as more effective to be used as biostimulants, significantly affecting crop growth performance, leaf gas exchange, WUE, and biochemical accumulation. The result showed that the integration of SiO_3_^2−^ compounds with BS significantly raised the rates of Ps, stomatal conductance, and transpiration rate, compared with BT and 0B. Treatment of BS applied on Si-treated plants significantly increased in Chl_a+b_ at higher rates, compared with BT or 0BS. The inoculation of the soil bacterium BS together with SiO_3_^2−^ compounds significantly increased Chl_a+b_ content and photosynthetic efficiency, which resulted in higher biomass production [18]. Increased Chl_a+b_ content in the leaf tissues denoted a favourable response to the accumulation of photosynthetic pigment, which is important for boosting plant growth under biotic stress conditions [19,20].

Jabborova et al. (2021) [21] noted that BS inoculation of soil-planting media led to increased Chl_a+b_ content and transpiration rate, which improved the growth performance of ginger. Furthermore, phytoene overaccumulation in the leaf chloroplasts was closely associated with a reduction in the Ps rate due to the impairment in chloroplast development and functionality, especially the plant grown under various environmental stress conditions [22]. Thus, the integration of SiO_3_^2−^ compounds with antagonistic bacteria attributed to the more efficient net assimilation rate and other physiological processes for admitting water, nutrients, and CO_2_ in and out of the plant system for better plant growth functions. In the present study, WUE was found to have declined when bananas were infected with *Fusarium* wilt disease. Moreover, the instantaneous WUE of the infected plant was decreased under environmental stress conditions due to the reduction in WUE to high turgor pressure in the plant [23,24]. Fernandes et al. (2020) [25] revealed that SiO_3_^2−^ compounds applied on infected black pepper plants significantly improved WUE and instantaneous WUE for mediating plant defence. According to [26], leaf thickness is a very important feature for CO_2_ assimilation, water saving, and nutrient availability. In addition, higher SLA values indicate high resource acquisition, better photosynthetic capacities, and higher plant growth rates. Quiroga et al. (2019) [27] noticed that lower SLA values showed thicker leaves associated with higher leaf protein concentration and better improvement in the photosynthetic capacity for plant growth functions. Greater canopy light interception improved plant performance and leaf gas exchange in terms of stomata opening that controls both the entry of CO_2_ and the water loss through the rate of transpiration to the atmosphere [28,29].

The production of other phytohormones and biochemical accumulation were triggered and affected the physiological responses, including the decrease in the rate of Ps, stomatal conductance and transpiration rate, when plants were exposed to various environmental stress conditions [30,31]. Similar results were found by [32], who reported the reduction in Ps rate due to the effects of fusaric acid on the photosynthetic activity accompanied by an increase in stomata closure which restricted water loss through the transpiration process in the infected tomato plant. According to [33,34], the infected plant usually develops secondary effects through the disturbance function of the photosystems that are located in the chloroplasts. Interestingly, the application of SiO_3_^2−^ compounds on soil planting media had significant suppression effects on microbial population in the soil at the end of the experiment. Moreover, BS inoculation on the root of Si-treated plant greatly suppressed FOC based on reduction in DI. Overall, the microbial number significantly increased with increased bacteria population due to inoculation of antagonistic bacteria for controlling *Fusarium* disease wilt in bananas. This brought a positive effect on plant disease resistance itself.

Kaushik and Saini (2019) [35] stated that soil application by integration of SiO_3_^2−^ compounds with antagonistic bacteria is recommended as a good strategy for controlling disease in vegetable crop production. The amount of TPC started to increase from 81.52 to 175.15 µg/g fresh weight when the proline content of the Berangan banana plant started to increase from 23.79 to 51.34 µmolg^−1^ fresh weight under different treatments condition (Figure 10). There was a strong significance with positive results, suggesting that an increase in the accumulation of TPC is directly proportional to proline content in banana plant tissues. Constant disease suppression management leads to a good reduction in biotic stress as well as proline content accumulation; however, an increase in TPC is also a good result for plant defence and better growth performance of bananas. In addition, soil application with effective microbes enhanced the biochemical defence responses of cucumber plants for controlling *Fusarium* root and stem rot [36]. The results showed that proline, TPC, TFC, and lignin content in plant tissues significantly increased when the root of the banana was infected with FOC. According to [37], the biochemical accumulation of proline, TFC, and TPC significantly increased, which contributed to the reduced root streak and leaf wilting observed in infected plants. Dong et al. (2020) [38] reported that the lignin content of banana roots also began to accumulate at the early infection stage in order to prevent serious infection from FOC.

In the present study, the results showed that root-to-shoot ratio, total leaf area, and total dry biomass of banana had synergistic interactions with the increasing application volume of SiO_3_^2−^ compound with BS inoculation. Ahmed et al. (2020) [39] mentioned that different Si rates and application levels have a significant impact on the growth performance and development of the wheat plant. With respect to the growth development, *Fusarium* wilt diseases seem to have serious side effects on the whole banana plant system and physiological traits, especially on the damage to the cell membrane and loss of integrity in addition to the produced reactive oxygen species to counteract disease infection [40,41]. Si application promoted growth, increased the efficiency of physiological processes as well as increased resistance of plants to nutrient stress. Si application was found to be involved in the modification of root growth under environmental stress conditions to provide better strength on plant tissue and improved root hydraulic conductance [42,43]. 

Ahammed and Yang (2021) [44] noticed that Si application on the root induced root silicification and improved mechanical properties as a physical barrier for restricting the penetration of the mycelia of the fungus. In other words, roots and stem at the base underground level were much thicker and more rigid following the application of SiO_3_^2−^ compounds [45]. Meanwhile, an increase in plant age with leaf number concurrently enhanced higher total leaf area, apart from allowing more light penetration into the plant canopy, which may influence the Ps rate efficiency and crop growth performance [41,46]. The results showed that there was a significant (*p* < 0.05) increase in RGR at the early growth stage in healthy samples. A similar trend was noted for the infected samples, but a significant reduction with increasing periods of growth was seen when samples were infected with fungal disease, compared with healthy samples. A similar result was found by [47], that a decreased RGR with increasing periods of growth was due primarily to the plant growth and development of *Telfairia occidentalis* apparently responding to an infection by a virus and the biotic stress conditions. The uptake of Si by the plant in terms of H_4_SiO_4_ was via the xylem from roots to shoots, but the amount of H_4_SiO_4_ uptake was greater in the roots compared with the shoot tissues because of the direct loss during a translocation process [48,49,50]. Shabbir et al. (2020) [51] reported that inoculation of *Enterobacter* sp. and *Arbuscular mycorrhizal* fungi with SiO_3_^2−^ compounds significantly improved Si content in the roots and shoots as well as leaf nutrient contents of rubber plants. However, the banana plant received only a small amount of H_4_SiO_4_, but its beneficial effects on RGR enhanced the assimilation rate production stimulated by a higher application volume of the SiO_3_^2−^ compound [52,53]. 

The most crucial growth rate performance that is closely related to plant development until harvest time is plant height [54]. Thus, the plant height of the banana is an indicator determining the optimum application volume of the SiO_3_^2−^ compound that is suitable to be applied. Regression analysis showed a polynomial trend between the plant height of the banana and the volume of K_2_SiO_3_ integrated with BS (Figure 11). Based on the result, the optimum volume of K_2_SiO_3_ was determined as 28.86 mL per plant. It seemed that soil application with K_2_SiO_3_ up to 28.86 mL with antagonistic bacteria significantly increased the plant height up to 27.91 cm; however, the plant growth performance of banana was reduced when the application volume applied more than the optimum volume. The regression analysis in Figure 12 showed that the optimum volume of Na_2_SiO_3_ was determined as 17.36 mL per plant suitable to be integrated with antagonistic bacteria. It seemed that application with the optimum amount of Na_2_SiO_3_ with antagonistic bacteria significantly increased the plant height to the maximum value of 28.14 cm under the *Fusarium* stress condition. 

## 4. Materials and Methods

### 4.1. Experimental Materials Preparation and Treatment

Both of the two separate experiments were conducted in a greenhouse at the University of Putra Malaysia (UPM), Selangor. Two weeks old Berangan banana seedlings were received from NNS Permata Holding, established in Kuala Pilah, Negeri Sembilan. Oxisol soil (Munchong series soil) was used as planting media that had clay texture (62.7% clay, 10.89% silt, and 26.21% sand) in 15 cm × 15 cm polybags. This soil type is commonly used for growing bananas in Malaysia, thus also used to test the plant growth performance and plant response against *Fusarium* wilt disease. Soil nutrient status was 1.27% organic carbon, 0.20% total N, 0.0032% available P, and 0.0057% available K.

All treatments for both experiments were arranged in a split-plot system in a randomized complete block (RCBD) design with four replications. In the first experiment, treatments in the main plots were divided into healthy plants without FOC inoculation treated with 25 mL of K_2_SiO_3_ solution and considered negative control (−ve FOC), and diseased plants treated with 25 mL of Na_2_SiO_3_ solution considered as a positive control (+ve FOC). Both SiO_3_^2−^ compounds were prepared at a constant concentration of 1% (w/v) as recommended by the Department of Agriculture (DOA), which was applied on soil planting media as well as the root of banana at 15 days interval (15DI). Different species of antagonistic bacteria were assigned as subplots; 0B (without *Bacillus* spp.) served as a control, BS (*Bacillus subtilis*) and BT (*Bacillus thuringiensis*). A 25 mL plant^−1^ (10^8^ CFU mL^−1^) of the antagonistic bacteria suspension was applied on soil planting media as well as the roots of banana seedlings following the treatment at 15DI. A similar experimental setup was used for the second experiment, but in sub-subplots; they were further divided into four different application volumes of SiO_3_^2−^ compounds (0, 20, 40, 60 mL) and integrated with the best selected antagonistic bacteria (BS) at 15DI (7.00 am to 8.30 am). All banana farm practices were followed, and applied procedures were based on the recommendations from the DOA.

Pure inoculum fungus of FOC-TR4 suspensions was received from the culture collection of the Department of Plant Protection, Faculty of Agriculture, UPM. Then, it was cultured and quantified by using a hemocytometer. Before transplanting, the roots of the banana were force inoculated with FOC-TR4 by immersing them for 30 min in a solution containing 10^6^ FOC spores per mL, while plantlet roots treated with distilled water served as a control. Again, soil planting media around the roots of the seedling was inoculated with 40 mL (10^8^ FOC spores mL^−1^) at the time of transplanting, following the soil inoculation procedure described by Slattery et al. [46]. However, only a healthy seedling with uniform size was transplanted into the polybag after being inoculated in the soil.

### 4.2. Data Collection

#### 4.2.1. Determination of Total Chlorophyll Content and Leaf Gas Exchange

The amount of total chlorophyll (Chl_a+b_) was determined by following the procedure outlined by Abro et al. [55]. Four discs were collected from the middle part of a banana leaf using a cork borer and placed into a plastic vial covered in aluminium foil with 20 mL of 80% acetone. To remove all of the pigments, the samples were stored in the dark for 7 days. A spectrophotometer (UV-3101PC UV-VIS-NIR, Shimadzu, Japan) was used, and the absorbance values of the solution of each sample were read at 647 nm and 664 nm. The amount of Chl_a=b_ contents was calculated as follows: Chl_a_ = 13.19 (A_664_) − 2.57 (A_647_)(1)
Chl_b_ = 22.1 (A_647_) − 5.26 (A_664_)(2)
Chl_a+b_ = 3.5 (Chl_a_ + Chl_b_)/4(3)
where, Chl_a_, Chl_b_, and Chl_a+b_ represent chlorophyll a, chlorophyll b, and total chlorophyll (a+b), respectively. A_647_ and A_664_ denote the absorbance of the solution at 647 and 664 nm, respectively, while 13.19, 2.57, 22.1, and 5.26 are the absorption coefficients, 3.5 (mL) was the total volume used in the analysis taken from the original solution, and 4 (cm^2^) was the total discs area. 

A portable photosynthesis system (Model: L1-6400, Li-COR Inc., Lincoln, NE, USA) was used to determine leaf gas exchange parameters. At 6th WAT, the third fully expanded leaves from each treatment were selected for determining the rate of photosynthesis (Ps), stomatal conductance and transpiration rate. The measurement was made at the standard time (0800 to 1100 am) by clipping the leaf on the chamber. However, measurements used optimal conditions set at 400 µmol mol^−1^ CO_2_, 30 °C cuvette temperature and 60% relative humidity with an airflow rate set at 500 cm^3^ min^−1^. The reading of Ps, stomatal conductance, and transpiration rate were expressed as µmol CO_2_ m^−2^s^−1^, mmol m^−2^s^−1^, and mmol H_2_O m^−2^s^−1^, respectively.

#### 4.2.2. Physiological Attributes

WUE is the ratio of the rate of carbon assimilation (Ps) to the rate of transpiration. The results were expressed in µmolCO_2_ assimilate mol^−1^ H_2_O loss.
WUE = Ps/transpiration rate
Int-WUE = Ps/stomatal conductance
where A = Ps (µmol CO_2_ m^−2^s^−1^), stomatal conductance (mmol m^−2^s^−1^), and transpiration rate (mol H_2_O m^−2^s^−1^).

#### 4.2.3. Biochemical Assay

TFC (Total flavonoid content) in the root sample was determined by the following method from Anthony et al. [56] with some modifications. About 5 g of root samples were grounded with liquid nitrogen, and the samples were mixed with 20 mL of 80% methanol added into a Falcon tube and placed in a shaking incubator for 30 min before being centrifuged for 15 min. Then, the supernatants of the centrifuged sample were filtered using Whatman No.1 filter paper. The extract was used for further analysis of antioxidant activity. The amount of TFC in the sample was measured using the colourimetric aluminium chloride method. In a test tube, 0.25 mL of the extract or catechin standard solution was combined with 1.25 mL of distilled water. Then, about 75 µL of 5% sodium nitrite solution and 150 µL of a 10% aluminium chloride solution were added after 6 min. After another 5 min, the mixture was given time to stand before receiving 0.5 mL of 1 M sodium hydroxide. The mixture was brought to 2.5 mL with distilled water and mixed well. The absorbance was measured immediately at 510 nm using a spectrophotometer (Model: Shimadzu UV-160A Visible Recording Spectrophotometer, Kyoto, Japan). TFC results were expressed as catechin equivalent (CE) per 100 mL sample. TPC (Total phenolic content) from root samples was determined by following the method from Rana et al. [57], and lignin content was measured by following the method described by Mansora et al. [58]. The results of TPC were expressed as μg gallic acid/g fresh weight, and lignin content was expressed as μg LTGA/g fresh weight. 

Determination of proline content in banana leaves was conducted according to the method described by Bates et al. [59] and expressed as µmolg^−1^ of fresh weight. Fresh banana leaves (0.5 g) were grounded in liquid nitrogen. Then, 10 mL of a 3% aqueous sulfosalicylic acid solution were added to the samples and filtered through Whatman paper (No. 2). About 2 mL of acetic acid and 2 mL of acidic ninhydrin reagent were added to a 2 mL aliquot. The mixture was thoroughly stirred and incubated in a boiling water bath for 1 h. Then, it was brought to room temperature in the ice bath. After the addition of 4 millilitres (mL), a spectrophotometer (Model: Shimadzu UV-160A Visible Recording Spectrophotometer, Shimadzu, Kyoto, Japan) was used to detect the extinction of the mixture’s upper toluene level at 518 nm. The amount of proline content was calculated as follows: Proline = [(µg proline mL^−1^ × 10 mL toluene)/115.5 µg µmole^−1^)]/(0.5 g sample/5)(4)
where 10 mL = volume of toluene; 115.5 µg µmole^−1^ = molecular weight of proline; 0.5 g = weight of fresh sample and 5 = dilution factor.

#### 4.2.4. Disease Assessment and Microbial Population

The number of banana plants exhibiting visible symptoms, such as chlorosis and necrosis of the leaves, was counted for each treatment to determine the disease incidence (DI) [60]. The following formula was adopted to calculate the percentage of DI: DI (%) = (number of infected seedlings/total number of seedlings assessed) × 100(5)

Microbial populations (fungal and bacterial) were quantified following the serial dilution and spread plate method described by Nwadibe et al. [61]. Microbial population for fungus and bacteria were quantified during the initial (1st week after transplanting) and final week (12th WAT) of the experimental period.

#### 4.2.5. Crop Growth Performance

The leaf area for each banana plant was calculated using the formula with a correction factor of 0.755 following [62] method as follows: Leaf area = [leaf length × maximum width × 0.755]. Total dry biomass was determined every 3 weeks after seedlings of banana were treated with different application volumes of SiO_3_^2−^ compounds. The root-to-shoot ratio was determined by the partitioning of dry matter of plant and calculated by the following formula [63]:Root to shoot ratio = total root dry weight (g)/total shoot dry weight (g)

The specific leaf area (SLA) was determined by dividing the area of all harvested leaves by the dry weight of the leaves [64]. The data of SLA were expressed in cm^2^ per dry weight leaves basis (cm^2^g^−1^), respectively. Data were calculated every 3 weeks after treatments by following the formula:SLA = total leaf area (cm^2^)/total dry weight (g)

The relative growth rate (RGR) is the growth rate relative to size [65]. RGR is a measure used to quantify the acceleration of plant growth performance. It is measured as the mass increase per aboveground biomass per time. RGR was calculated by using the formula: RGR = LnW2-LnW1/(t2 − t1)
where W1 = dry weight of plant m-2 recorded at time t1 (g), W2 = Dry weight of plant m^−2^ recorded at time t2 (g), and t2 − t1 = interval time (month).

### 4.3. Statistical Analysis

All collected data from these experiments were analyzed using analysis of variance (ANOVA) by a statistical analysis system (SAS 9.4) to decide the significance of the effect among treatments. Differences between means separated were made using the least significant difference (LSD) at *p* < 0.05.

## 5. Conclusions

*Fusarium* wilt disease negatively affected the plant growth and physiological parameters of the banana. The application of biostimulant Si-based compounds integrated with BS inoculation on planting media significantly increased plant growth and a more robust response against the invading pathogen to reduce *Fusarium* wilt incidence in bananas. From these results, it was found that the biostimulant activity of SiO_3_^2−^ compound and antagonistic bacteria enhanced the physiological growth parameters and biochemical changes for disease suppression. Based on the presented result, the disease suppression significantly reduced the accumulation of proline content but increased the TPC. The amount of TPC started to increase from 81.52 to 175.15 µg/g fresh weight when the proline content of the banana plant started to increase from 23.79 to 51.34 µmolg^−1^ fresh weight under different treatments condition. Moreover, the integration with optimum K_2_SiO_3_ application volume of 28.86 mL with BS inoculation enhanced the plant height of banana by 27.91 cm, whereas Na_2_SiO_3_ integrated with BS inoculation on soil planting media significantly reduced the *Fusarium* wilt incidence of banana by 56.25% and significantly increased the major physiological growth characteristics. Therefore, it was recommended that infected bananas should be treated with 17.36 mL of Na_2_SiO_3_ with BS to stimulate better plant growth performance. In the future, field trial experiment is recommended, and further studies should be focused on the effectiveness of biostimulant activity of SiO_3_^2−^ compounds and BS inoculation able to suppress *Fusarium* wilt disease in banana grown in flooded areas in order to ensure food security in face of global climate change scenarios.

## Figures and Tables

**Figure 1 plants-12-01124-f001:**
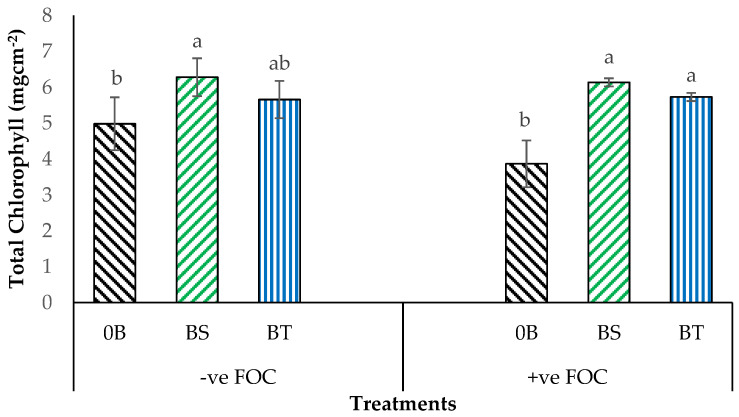
Significant interaction between different treatments on banana seedlings and antagonistic bacteria application on total chlorophyll (Chl_a+b_) content at 6th WAT. Data are mean ± SEM (standard error of differences between means) of 24 replicates. Bars represent means followed by the different letters in lowercase, significant at *p* < 0.05. Soil uninoculated with FOC (−ve FOC) served as negative control, and soil inoculated with FOC (+ve FOC) served as positive control. 0B—without *Bacillus* spp., BS—*B. subtilis*, and BT—*B. thuringiensis*.

**Figure 2 plants-12-01124-f002:**
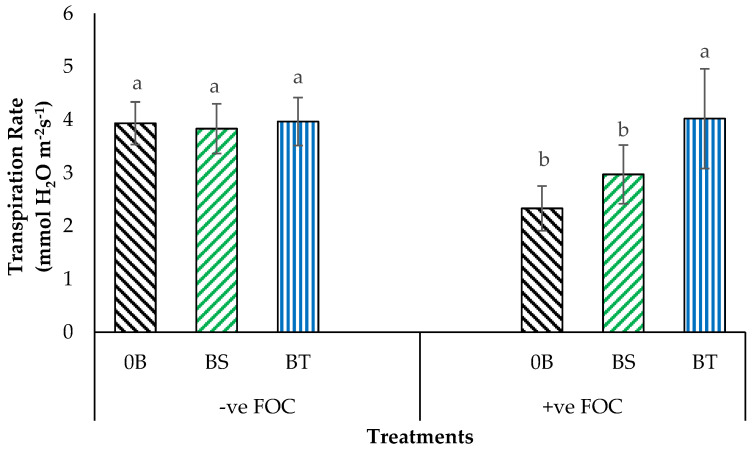
Significant interaction between different treatments of banana seedlings and application of antagonistic bacteria on transpiration rate at 6th WAT. Data are mean ± SEM (standard error of differences between means) of 24 replicates. Bars represent means followed by the different letters in lowercase significant at *p* < 0.05. Soil uninoculated with FOC (−ve FOC) served as negative control, and soil inoculated with FOC (+ve FOC) served as positive control. 0B—without *Bacillus* spp., BS—*B. subtilis*, and BT—*B. thuringiensis*.

**Figure 3 plants-12-01124-f003:**
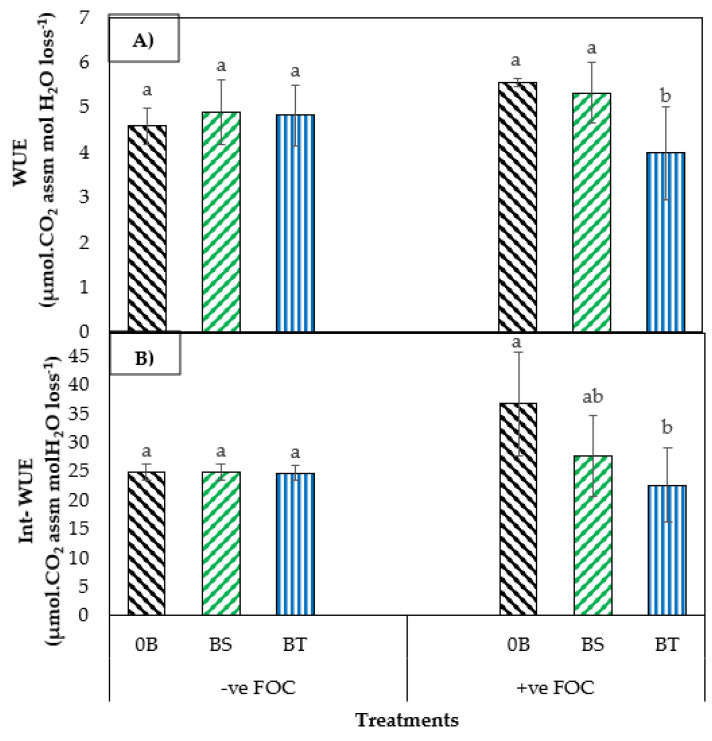
Effect of different treatments on banana seedlings and the application of antagonistic bacteria 6th WAT: (**A**) changes in water-use efficiency (WUE), and (**B**) changes in intrinsic-WUE (Int-WUE). Data are mean ± SEM (standard error of differences between means) of 24 replicates. Bars represent means followed by the different letters in lowercase, significant at *p* < 0.05. Soil uninoculated with FOC (−ve FOC) served as negative control, and soil inoculated with FOC (+ve FOC) served as positive control; 0B—without *Bacillus* spp., BS—*B. subtilis*, and BT—*B. thuringiensis*.

**Figure 4 plants-12-01124-f004:**
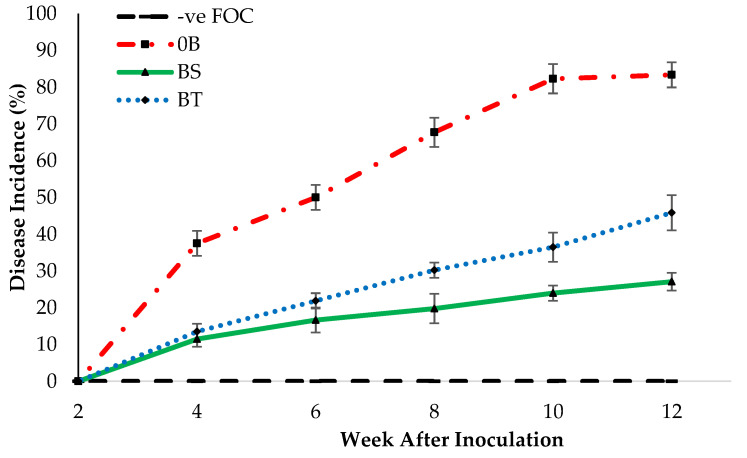
Disease incidence (DI) as affected by different treatments on banana seedlings and antagonistic bacteria application throughout the 12th WAT. 0B—without *Bacillus* spp., BS—*Bacillus subtilis*, and BT—*Bacillus thuringiensis*.

**Figure 5 plants-12-01124-f005:**
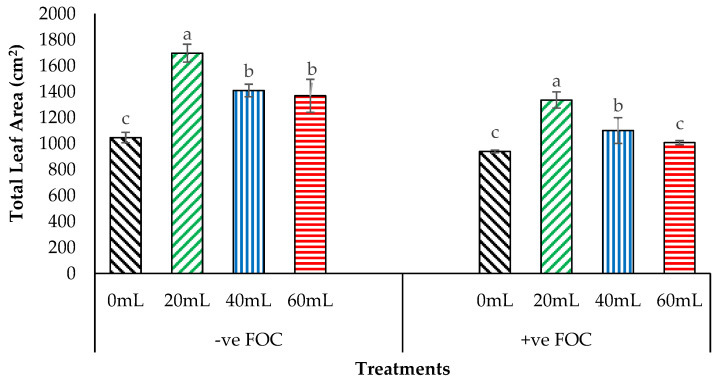
Significant interactions between different treatments on banana seedlings and application volume of SiO_3_^2−^ compounds on the total leaf area 9th WAT. Data are mean ± SEM (standard error of differences between means) of 32 replicates. Bars represent means followed by the different letters in lowercase significant at *p* < 0.05. Soil uninoculated FOC (−ve FOC) served as negative control, and soil inoculated FOC (+ve FOC) served as positive control.

**Figure 6 plants-12-01124-f006:**
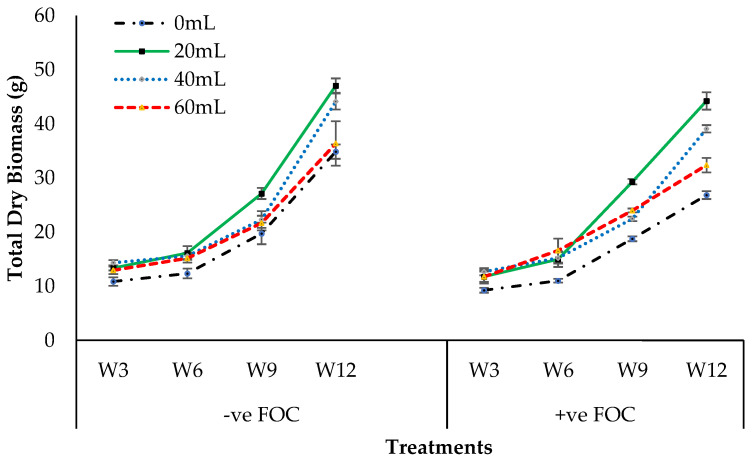
Effect of different treatments of banana seedlings and application volume of SiO_3_^2−^ compounds throughout the 12th WAT on the total dry biomass. Data are mean ± SEM (standard error of differences between means) by least significant difference (*p* < 0.05). Soil uninoculated with FOC (−ve FOC) served as negative control, and soil inoculated with FOC (+ve FOC) served as positive control.

**Figure 7 plants-12-01124-f007:**
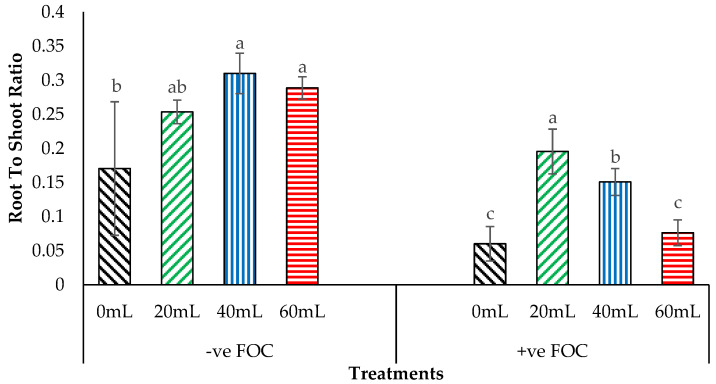
Significant interactions between different treatments of banana seedlings and application volume of SiO_3_^2−^ compounds on the root-to-shoot ratio 9th WAT. Data are mean ± SEM (standard error of differences between means) of 32 replicates. Bars represent means followed by the different letters in lowercase, significant at *p* < 0.05. Soil uninoculated with FOC (−ve FOC) served as negative control, and soil inoculated with FOC (+ve FOC) served as positive control.

**Figure 8 plants-12-01124-f008:**
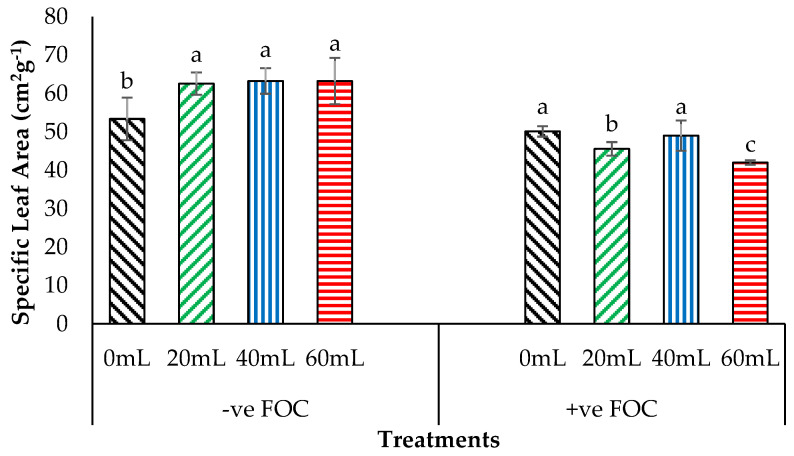
Significant interaction between different treatments of banana seedlings and the application volume of SiO_3_^2−^ compounds on specific leaf area (SLA) 9th WAT. Data are mean ± SEM (standard error of differences between means) of 32 replicates. Bars represent means followed by the different letters in lowercase, significant at *p* < 0.05. Soil uninoculated with FOC (−ve FOC) served as negative control, and soil inoculated with FOC (+ve FOC) served as positive control.

**Figure 9 plants-12-01124-f009:**
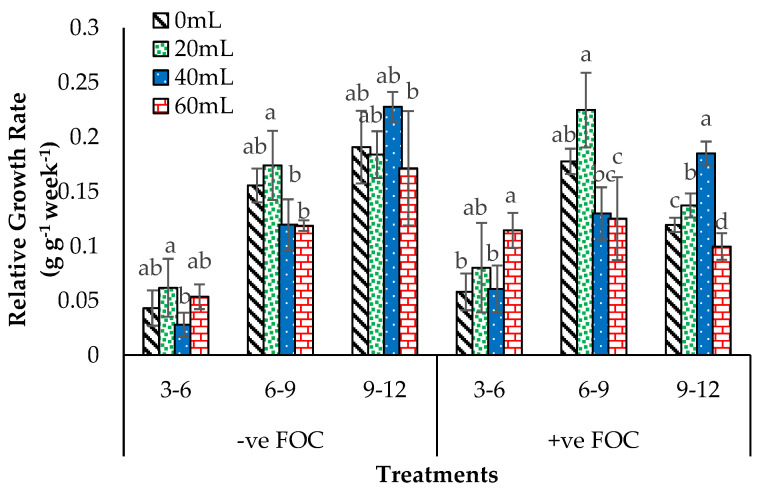
Effect of different treatments of the banana seedlings and the application volume of SiO_3_^2−^ compounds throughout the 12th WAT on relative growth rate (RGR). Data are mean ± SEM (standard error of differences between means) of 32 replicates. Bars represent means followed by the different letters in lowercase significant at *p* < 0.05. Soil uninoculated with FOC (−ve FOC) served as negative control, and soil inoculated with FOC (+ve FOC) served as positive control.

**Figure 10 plants-12-01124-f010:**
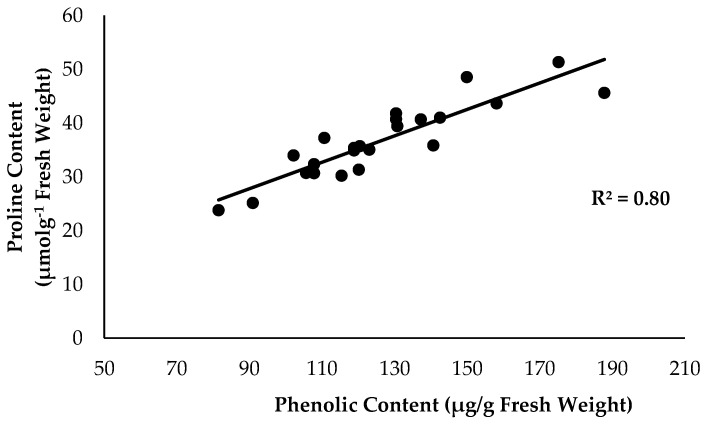
Relationship between total phenolic content (TPC) and proline content as influenced by integration with different treatments of banana and antagonistic bacteria inoculation.

**Figure 11 plants-12-01124-f011:**
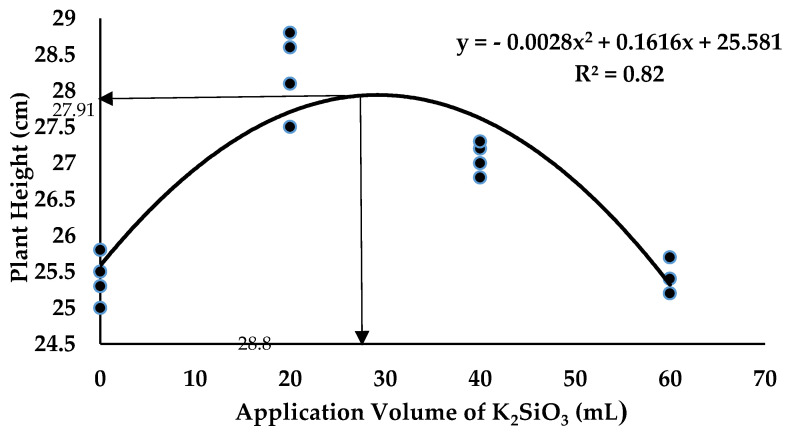
Regression analysis between plant height of banana and application volume of K_2_SiO_3_ integrated with BS inoculation 12th WAT.

**Figure 12 plants-12-01124-f012:**
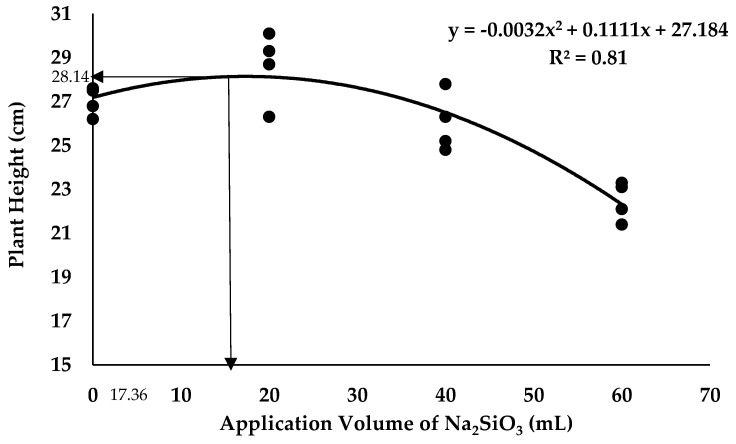
Regression analysis between plant height of banana and application volume of Na_2_SiO_3_ integrated with BS inoculation 12th WAT.

**Table 1 plants-12-01124-t001:** Photosynthesis rate (Ps) and stomatal conductance as affected by different treatments on banana seedlings and antagonistic bacteria application at 6th WAT.

Factors	Photosynthesis Rate	Stomatal Conductance
	(µmol CO_2_ m^−2^s^−1^)	(mmol m^−2^s^−1^)
Main plot means: Treatments		
−ve FOC	18.43 ± 0.58 ^a^	0.74 ± 0.04 ^a^
+ve FOC	14.59 ± 1.89 ^b^	0.56 ± 0.23 ^a^
LSD (*p* < 0.05)	0.98 **	NS
Subplot means: Antagonistic bacteria		
Control (0B)	15.41 ± 3.11 ^b^	0.55 ± 0.21 ^b^
*B. subtilis* (BS)	17.03 ± 1.75 ^a^	0.67 ± 0.17 ^ab^
*B. thuringiensis* (BT)	17.08 ± 1.99 ^a^	0.74 ± 0.13 ^a^
LSD (*p* < 0.05)	1.37 *	0.14 *
Significance interaction	NS	NS

Means followed by the same letter within a column are not significantly different at *p* > 0.05 by least significant difference (LSD) with n = 24; * and ** are significant differences at *p* < 0.05 and 0.01, respectively. NS—not significant. FOC u-inoculated soil (−ve FOC) served as negative control, and soil FOC—inoculated soil (+ve FOC) served as positive control. 0B—without *Bacillus* spp., BS—*B. subtilis*, and BT—*B. thuringiensis*.

**Table 2 plants-12-01124-t002:** Leaf proline content, root total flavonoid content (TFC), root total phenolic content (TPC), and root lignin content in plant tissues of banana seedlings affected by different treatments and antagonistic bacteria application 12th WAT.

Factors	Leaf Proline Content	Total Flavonoid Content	Total Phenolic Content	Lignin Content
	(µmolg^−1^ Fresh Weight)	(mg CE/100 mL)	(µg/g FW)	(LTGAg^−1^ Tissue)
Main plot means: Treatments				
−ve FOC	32.65 ± 5.29 ^b^	8.02 ± 1.62 ^b^	110.17 ± 14.53 ^b^	1.32 ± 0.65 ^a^
+ve FOC	40.72 ± 5.61 ^a^	11.20 ± 2.46 ^a^	142.26 ± 22.14 ^a^	1.76 ± 0.43 ^a^
LSD (*p* < 0.05)	2.01 **	1.97 *	17.98 *	0.56 *
Subplot means: Antagonistic bacteria				
Control (0B)	36.82 ± 5.59 ^a^	11.62 ± 2.35 ^a^	128.48 ± 15.95 ^a^	1.09 ± 0.43 ^b^
*B. subtilis* (BS)	32.17 ± 4.89 ^b^	7.31 ± 1.65 ^c^	110.67 ± 19.03 ^b^	1.89 ± 0.54 ^a^
*B. thuringiensis* (BS)	41.07 ± 7.01 ^a^	9.90 ± 1.81 ^b^	139.5 ± 29.75 ^a^	1.65 ± 0.50 ^a^
LSD (*p* < 0.05)	4.59 **	0.73 ***	12.84 **	0.51 *
Significance interaction	NS	NS	NS	NS

Means followed by the same letter within a column do not represent the significant difference *p* > 0.05 by least significant difference (LSD) with n = 24. *, **, and *** are significant difference at *p* < 0.05, 0.01, and 0.001, respectively; NS—not significant. Soil uninoculated with FOC (−ve FOC) served as negative control, and FOC-inoculated soil (+ve FOC) served as positive control. 0B—without *Bacillus* spp., BS—*B. subtilis*, and BT—*B. thuringiensis*.

**Table 3 plants-12-01124-t003:** Effect of different treatments on banana seedling and antagonistic bacteria application on soil microbial population.

Factors	Initial Experiment (CFU log_10_ g^−1^)		Final Experiment (CFU log_10_ g^−1^)	
	Fungus	Bacteria	Fungus	Bacteria
Main-plot means: Treatments				
−ve FOC	4.10 ± 0.19 ^a^	4.04 ± 0.44 ^a^	3.99 ± 0.18 ^b^	4.03 ± 0.52 ^a^
+ve FOC	4.12 ± 0.25 ^a^	4.19 ± 0.09 ^a^	4.12 ± 0.20 ^a^	3.86 ± 0.61 ^a^
LSD (*p* < 0.05)	NS	NS	0.04 **	NS
Subplot means: Antagonistic bacteria				
Control (0B)	4.18 ± 0.14 ^a^	4.20 ± 0.09 ^a^	4.22 ± 0.13 ^a^	3.46 ± 0.77 ^b^
*B. subtilis* (BS)	4.01 ± 0.19 ^a^	4.16 ± 0.06 ^a^	3.90 ± 0.20 ^b^	4.24 ± 0.15 ^a^
*B. thuringiensis* (BS)	4.15 ± 0.28 ^a^	3.98 ± 0.54 ^a^	4.04 ± 0.14 ^b^	4.14 ± 0.13 ^a^
LSD (*p* < 0.05)	NS	NS	0.14 **	0.55 *
Significance interaction	NS	NS	NS	NS

Means followed by the same letter within a column do not represent significant difference at *p* > 0.05 by least significant difference (LSD) with n = 24. * and ** represent significant difference at *p* < 0.05 and 0.01, respectively; NS—not significant. Soil uninoculated with FOC (−ve FOC) served as negative control, and FOC- inoculated soil (+ve FOC) served as positive control. 0B—without *Bacillus* spp., BS—*B. subtilis*, and BT—*B. thuringiensis*.

## Data Availability

Not applicable.

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
