# Peer review of "Biostimulant Activity of Silicate Compounds and Antagonistic Bacteria on Physiological Growth Enhancement and Resistance of Banana to Fusarium Wilt Disease"

_plants, 2023, doi:10.3390/plants12051124_

Round 1

Reviewer 1 Report

Summary

This study collected plentiful meaningful data about the effects of silicates and Bacillus spp. on banana seedlings with Fusarium wilt disease. However, I have a few questions on the experimental design mentioned in my general comments. English language needs improvement. I recommend authors find colleagues who are proficient in English to review this article. Some small typos can be found in my specific comments.

General comments

1.       ‘K2SiO3 applied on soil uninoculated FOC and Na2SiO3 applied on soil inoculated FOC’ Why not applying the same silicate? Will this difference affect the results?

2.       Methods description was not clear. In line 516, ‘Si compounds’ here refers to K2SiO3 and Na2SiO3? The concentration is w/w or w/v, and unit? How much silicate solution was applied at each time? Similar questions on the Bacillus applications, how much bacillus sp. was applied in each treatment? The ‘2 WAT at 15 DI’ and ’15 DI’ refer to only the bacillus sp. application or both the silicate solution and bacillus sp. applications?

3.       How many plants/seedlings were included in each treatment?

4.       Bars and lines in all figures were not clear. Since different colors were used, pattern fill seems not necessary. However, if authors were trying to make the figure accessible to audiences with color blindness, I would recommend authors add boarder to the bars to make them a little clearer.

5.       Figure 1 & line 80: ‘Significant interaction’ means the effects of Bacillus spp. and FOC are interacted. However, I didn’t find such comparation in results. Besides, did you compare total chlorophyll between -ve FOC and +ve FOC? I don’t understand why you set a negative control and a positive control. If both of them are control, where are the other treatments?

6.       It is not rigorous to express ‘Integration of Na2SiO3 significantly suppressed FOC’ in Line 89. Chlorophyll a & b can not be used as direct parameters to estimate the colonization of FOC. Similar problems can be found in abstract and conclusions. In this study, silicate was applied before or the same time of FOC infection on banana seedlings. However, in abstract and conclusions, authors recommend FOC infected bananas can be treated with silicate. That are different. As scientific research manuscript authors, we should focus on what we did and express it precisely.

Specific comments

Line 40: ‘According to Wong et al. [3],’ similar correction is need in line 43, 60, 532, 538, 574, 593, 596

Line 54: looking for instead of looking forward

Lines 66-68: English can be improved, similar problem in lines 54-55.

Line 87: weeks

Line 511: Please uniform the units.

Line 516: Si compounds here refers to K2SiO3 and Na2SiO3? The concentration is w/w or w/v, and unit? How much silicate solution was applied at each time?

Line 520: I recommend authors explain the abbreviations again the first time they appeared in materials and methods. I am one of the audiences prefer reading materials and methods earlier than reading results and discussions. For example, ‘WAT’. 2096992

Line 521: levels

Line 532: Do you have selection criteria for the transplanted seedlings?

Line 575: grounded, similar in line 597.

Line 576: 20 mL

Lines 575-581: The method was repeated.

Line 585: English language needed to be improved.

Line 591: Why not use TFC here?

Lines 593, 594, 596: Why use FW in 593 & 594, but ‘fresh weight in line 596? I recommend authors use full names instead of unnecessary abbreviations in the manuscript. For example, if the one expression won’t appear too many times, we can use the full name. It is easier for readers to understand your work.

Line 623: dry biomass. After what treatment?

Line 624: by instead of to

Line 640: How often RGR was measured? I am confusing about the formula to calculate RGR, do authors mean dry weight of aboveground biomass? If yes, you have to cut the whole plant, then how did you continue your experiment? Why use logarithmic transformation here?

Line 647: delete ‘by’, ‘an’

Line 649: ANOVA is used to decide the significance of effect among treatments.

Lines 658-660: Please improve English language.

Line 124: ‘letters in lower case’ instead of ‘small letters’

Reviewer 2 Report

The paper "Biostimulant Activity of Silicate Compounds and Antagonistic Bacteria on Physiological Growth Enhancement and Resistance of Banana to Fusarium Wilt Disease" aims to study the optimal doses of two silicates, K2SiO3 and Na2SiO3, applied on soil to protect the plant against pathogens.

Starting from the Abstract, the writing style is missing coherence and important words, like predicates.

The Introduction does not provide enough background and motivation for the study, it is too short, with only 11 not-so-relevant references, therefore it should be enriched and focused on innovative approaches.

The Methodology requires more details on the reasons why "Potassium silicate (K2SiO3) applied on soil uninoculated FOC and sodium silicate (Na2SiO3) applied on soil inoculated FOC before integrating with antagonistic bacteria", meaning the hypothetical different mechanisms of K2SiO3 vs Na2SiO3.

Going to section 4. Materials and Methods, particular aspects can be suggested, as follows:

L 503,535,536 etc: Please revise the sub-section number (4.1 , 4.2, 4.2.1 etc)

L538: The reference [55] is not related to the chlorophyll determination, please revise.

L564: "stomatal conductance" is preferred to "stomata conductance";

L574: TFC abbreviation (Total flavonoid content) can be defined here at the first appearance in sub-section.

L579-581 repeat the details in L575-578, please revise.

L592: Please define "TPC" abbreviation (probably Total polyphenols content).

L593: Ref. [57] does not refer to TPC or lignin determination, please provide the relevant reference.

L596: Ref. [58] does not detail the protocol for proline evaluation, instead it refers to Bates et al. (1973), please use the original reference or references that provide details of the protocols.

L598-601 and throughout the manuscript: Please use numbers for amounts instead of words (2mL,4mL). Also, adopt an uniform manner of units, for example mL, instead of ml;

L608: Please correct "toluence" to "toluene";

L613: Ref. [59] is a review that does not provide any detail on how to determine the disease incidence;

L617: Ref [60] does not describe the details of the spread plate method;

And i stop here with verifying the corresponding references because a major revision of the appropriate citations is more than necessary for the entire manuscript.

L661: Please revise the units µmg/g.

In the first part of the Results section (2.1,2.2,2.3), the influence of the treatment volumes of silicates is not discussed. Different influences of different treatments should be more clearly presented and argued.

The Discussion section should start from the main results and should more clearly discuss possible mechanisms and synergism, while grounding the hypotheses on similar findings.

Round 2

Reviewer 1 Report

I accept the revisions authors made in response to my comments. I would recommend editor to accept this edition.

Reviewer 2 Report

The paper still requires corrections and text editing.

The Abstract still misses predicates starting from the first phrase.

L68-70: Please revise: "...were effective, environmentally safe, and cost effective ... were developed [7]";

L72: Please delete "(Si)". Silicates are [SiO(4-x)](4-2x)-;

L115-117: Revise "was noticeably affected (p<0.05) with high significant interactions between treatments on the banana seedlings and antagonistic bacteria were observed."

L119-122: Same raw repeated, please revise;

L126-128: Please revise;

L135,146 are unreadable;

L148,149. Please revise;

L157-158: Please revise "in comparison to the negative control (-ve FOC + 0B) had the lowest stomatal conductance was 0.55 mmol m-2s-1."

L162-165: Please use shorter phrases;

L366-368: Please revise: "Figure 7 illustrate that the root to shoot ratio of Berangan banana seedlings were 40 mL of Na2SiO3 which had the highest (0.309), followed by 60 mL (0.288)..."

L439,440,441 are repeated, please revise;

L444-445: Revise: "Based on the result, the increase in rate of Ps attributed to the larger total leaf area and higher Chla+b in plant treated with enriched Si with BS."

L447-450: Please revise and use shorter phrases;

L453,662 are unreadable;

L454-462 are identical with L442-450, please delete L454-462;

L918-924: References should be revised again.

The writing style still requires extensive editing.
